# Optimization of Fenton Technology for Recalcitrant Compounds and Bacteria Inactivation

**Pablo Salgado [1,2]**, **José Luis Frontela [1] and Gladys Vidal [1,*]**

[1]  Grupo de Ingeniería y Biotecnología Ambiental, Facultad de Ciencias Ambientales y Centro EULA-Chile, Universidad de Concepción, Concepción 4070386, Chile; pablosalgado@udec.cl (P.S.); jfrontela2018@udec.cl (J.L.F.)
[2]  Departamento de Ingeniería Civil, Facultad de Ingeniería, Universidad Católica de la Santísima Concepción, Alonso de Ribera 2850, Concepción 4030000, Chile
[*]  Correspondence: glvidal@udec.cl; Tel.: +56-41-2204067

**Abstract:** In this work, the Fenton technology was applied to decolorize methylene blue (MB) and to inactivate *Escherichia coli* K12, used as recalcitrant compound and bacteria models respectively, in order to provide an approach into single and combinative effects of the main process variables influencing the Fenton technology. First, Box–Behnken design (BBD) was applied to evaluate and optimize the individual and interactive effects of three process parameters, namely $Fe^{2+}$ concentration ($6.0 \times 10^{-4}$, $8.0 \times 10^{-4}$ and $1.0 \times 10^{-3}$ mol/L), molar ratio between $H_2O_2$ and $Fe^{2+}$ (1:1, 2:1 and 3:1) and pH (3.0, 4.0 and 5.0) for Fenton technology. The responses studied in these models were the degree of MB decolorization ($D_\%^{MB}$), rate constant of MB decolorization ($k_{app}^{MB}$) and *E. coli* K12 inactivation in uLog units ($I_{uLog}^{EC}$). According to the results of analysis of variances all of the proposed models were adequate with a high regression coefficient ($R^2$ from 0.9911 to 0.9994). BBD results suggest that $[H_2O_2]/[Fe^{2+}]$ values had a significant effect only on $D_\%^{MB}$ response, $[Fe^{2+}]$ had a significant effect on all the responses, whereas pH had a significant effect on $D_\%^{MB}$ and $I_{uLog}^{EC}$. The optimum conditions obtained from response surface methodology for $D_\%^{MB}$ ($[H_2O_2]/[Fe^{2+}]$ = 2.9, $[Fe^{2+}]$ = $1.0 \times 10^{-3}$ mol/L and pH = 3.2), $k_{app}^{MB}$ ($[H_2O_2]/[Fe^{2+}]$ = 1.7, $[Fe^{2+}]$ = $1.0 \times 10^{-3}$ mol/L and PH = 3.7) and $I_{uLog}^{EC}$ ($[H_2O_2]/[Fe^{2+}]$ = 2.9, $[Fe^{2+}]$ = $7.6 \times 10^{-4}$ mol/L and pH= 3.2) were in good agreement with the values predicted by the model.

**Keywords:** recalcitrant compounds; *E. coli* K12; methylene blue; optimization; Pareto chart; perturbation graph

## 1. Introduction

Some of the effluents produced by industries such as textiles, dyes, tanneries, cosmetics and pulp are colored [1]. In the pulp industry, effluents are colored due to the presence of lignin byproducts and other phenolic compounds formed [2]. These compounds are considered dangerous and recalcitrant because of their low biodegradability and resistant to chemical degradation [1,3]. Besides, recalcitrant compounds with biological activity contained in treated effluent discharges are generating a loss of biodiversity in ecosystems. Even more, some of these compounds with benzyl and phenolic structures are considered endocrine disruptors [4–7]. In addition to the presence of recalcitrant compounds, the bacteria in the effluents of the pulp industry must also be seriously considered. The presence of bacteria in effluents discharged to water bodies are generating humans and animals diseases. In this sense, *Escherichia coli* and other bacteria have been identified in pulp industry effluent [8]. The presence of these bacteria in the effluents of the pulp industry raises an important concern regarding the current technologies (biological treatment) and regulations that govern the discharge of these

effluents [9]. The inactivation of a wide range of pathogens in the cellulose industry effluent is effective by chlorination at a relatively low cost [10,11]. However, despite its effectiveness there is a problem to consider: the formation of organochlorine compounds [12]. Tawabini, et al. [13] states that chlorine has a high reactivity that affects the formation of these byproducts (chlorinated organic compounds) when reacting with organic matter. These byproducts are characterized by high toxicity and mutagenic capacity for the environment.

The low effectiveness of conventional water treatments in the destruction of recalcitrant contaminants and the formation of hazardous byproducts has encouraged the search for treatments with a higher oxidative capacity avoiding the formation of harmful byproducts. In this sense, it has been proposed the use of the so-called advanced oxidation processes (AOP) for the elimination of recalcitrant compounds and disinfection of effluents [3,14,15]. Nevertheless, there are differences between bacterial inactivation and decolorization of recalcitrant organic compounds by AOP [16]. Bacteria with the ability to self-repair and grow again after damage are much more complex than recalcitrant compounds [17].

A common point of the vast majority of AOP is the formation of hydroxyl radicals ($\cdot$OH). Furthermore, $\cdot$OH is considered one of the species with the greatest oxidizing power. For example, chlorinated compounds used in conventional effluent treatments such as $Cl_2$ and $ClO_2$ have standard reduction potentials of 1.36 and 1.27 V/SHE respectively, while $\cdot$OH has a standard potential of 2.8 V/SHE [18].

Among the AOP, the Fenton technologies has focused a lot of attention for many years [19]. This process involves the reaction of $H_2O_2$ as an oxidant agent with $Fe^{2+}$ ions as a metal catalyst to produce the degradation agent of $\cdot$OH as illustrated in Equation (1). $Fe^{3+}$ produced by the Fenton reaction can also oxidize $H_2O_2$ to produce perhydroxyl radicals ($HO_2\cdot$; Equation (2)), named the Fenton-like reaction. The $\cdot$OH and $HO_2\cdot$ produced in Fenton and Fenton-like reactions can participate in parallel reactions to produce singlet oxygen ($^1O_2$; Equations (3) and (4)).

$$Fe^{2+} + H_2O_2 \rightarrow Fe^{3+} + OH^- + \cdot OH \tag{1}$$

$$Fe^{3+} + H_2O_2 \rightarrow Fe^{2+} + HO_2\cdot + H^+ \tag{2}$$

$$HO_2\cdot + HO_2\cdot \rightarrow H_2O_2 + {}^1O_2 \tag{3}$$

$$HO_2\cdot + \cdot OH \rightarrow H_2O + {}^1O_2 \tag{4}$$

Andreozzi, et al. [20] highlight the reactivity of $\cdot$OH, since this species has adequate properties to attack organic compounds, in addition to reacting $10^6$–$10^{12}$ times faster than other oxidants. Additionally, the cellular damage produced by $\cdot$OH in the disinfection processes takes place on different macromolecules present in the bacterial membrane causing its inactivation [21].

Many of the parameters that can affect any type of chemical reaction could affect the Fenton reaction, among which the effects of pH and reagent concentration stand out. The pH is one of the most important parameters in the Fenton reaction. However, it is possible to find that the optimum pH varies. One of the reasons why it is possible to find this variety at the optimum pH of the Fenton reaction may be associated with the speciation of $Fe^{2+}$ and $Fe^{3+}$ [22], changes in redox potentials of the main oxidizing species produced [23], or changes in the type of oxidizing species produced depending on the pH [24–26].

Without the presence of $Fe^{2+}$ in the Fenton system there is no formation of $\cdot$OH, so the presence of $Fe^{2+}$ is essential. However, it has been studied that too high $Fe^{2+}$ concentrations can cause the Fenton reaction oxidizing capacity to decrease (Equation (5)) [27].

$$\cdot OH + Fe^{2+} \rightarrow Fe^{3+} + OH^- \tag{5}$$

The $H_2O_2$ concentration, like $Fe^{2+}$, is also essential in Fenton systems [28]. However, an excess of $H_2O_2$ could act as a scavenger of ·OH [27,28] according to the Equation (6).

$$·OH + H_2O_2 \rightarrow HO_2· + H_2O \tag{6}$$

Accordingly, to minimize $Fe^{2+}$ and $H_2O_2$ acting as scavengers, but maximizing the production of oxidizing species from these reagents, it is very important to know the optimal $[H_2O_2]/[Fe^{2+}]$ [29].

Therefore, the aim of this work is to evaluate the decolorization of a model recalcitrant compound (methylene blue) and the inactivation of a model bacteria (*E. coli* K12 strain) by Fenton technology considering the operational parameters such as pH, $Fe^{2+}$ concentration ($[Fe^{2+}]$) and the molar ratio between $H_2O_2$ and $Fe^{2+}$ ($[H_2O_2]/[Fe^{2+}]$), and to reveal the single and combinative effects of the these variables influencing on degree of methylene blue (MB) decolorization ($D_\%^{MB}$), rate constant of MB decolorization ($k_{app}^{MB}$) and *E. coli* K12 inactivation in uLog units ($I_{uLog}^{EC}$).

## 2. Results and Discussion

### 2.1. Models and Regression Analysis

Table 1 lists the factors and levels in the experimental design using the following variable: pH, $Fe^{2+}$ concentration ($[Fe^{2+}]$, mol/L) and molar concentration ratio of $Fe^{2+}$ and $H_2O_2$ ($[H_2O_2]/[Fe^{2+}]$). Also, the experimental degree of MB decolorization ($D_\%^{MB}$), the apparent rate constant of MB decolorization ($k_{app}^{MB}$) and the inactivation of *E. coli* K12 bacteria in logarithmic units ($I_{uLog}^{EC}$) are presented.

**Table 1.** Actual values and coded levels (in parentheses) of the variables in the Box–Behnken design and experimental values for each response.

| | Variables | | | Responses (Experimental Results) | | |
|---|---|---|---|---|---|---|
| # | $[H_2O_2]/[Fe^{2+}]$ | $[Fe^{2+}]$ (mol/L) | pH | $D_\%^{MB}$ (%) | $k_{app}^{MB}$ (min$^{-1}$) | $I_{uLog}^{EC}$ (uLog) |
| 1 | 1:1 (−1) | $6.0 \times 10^{-4}$ (−1) | 4.0 (0) | 72.8 | 1.02 | 0.41 |
| 2 | 3:1 (+1) | $6.0 \times 10^{-4}$ (−1) | 4.0 (0) | 89.3 | 1.41 | 0.84 |
| 3 | 1:1 (−1) | $1.0 \times 10^{-3}$ (+1) | 4.0 (0) | 81.6 | 1.94 | 0.78 |
| 4 | 3:1 (+1) | $1.0 \times 10^{-3}$ (+1) | 4.0 (0) | 92.9 | 1.43 | 0.23 |
| 5 | 1:1 (−1) | $8.0 \times 10^{-4}$ (0) | 3.0 (−1) | 82.4 | 1.22 | 0.33 |
| 6 | 1:1 (−1) | $8.0 \times 10^{-4}$ (0) | 5.0 (+1) | 96.1 | 1.63 | 0.15 |
| 7 | 3:1 (+1) | $8.0 \times 10^{-4}$ (0) | 3.0 (−1) | 75.8 | 1.32 | 0.53 |
| 8 | 3:1 (+1) | $8.0 \times 10^{-4}$ (0) | 5.0 (+1) | 90.5 | 1.42 | 0.34 |
| 9 | 2:1 (0) | $6.0 \times 10^{-4}$ (−1) | 3.0 (−1) | 89.1 | 1.20 | 0.16 |
| 10 | 2:1 (0) | $1.0 \times 10^{-3}$ (+1) | 3.0 (−1) | 96.7 | 1.89 | 0.18 |
| 11 | 2:1 (0) | $6.0 \times 10^{-4}$ (−1) | 5.0 (+1) | 85.2 | 1.69 | 0.15 |
| 12 | 2:1 (0) | $1.0 \times 10^{-3}$ (+1) | 5.0 (+1) | 89.2 | 1.78 | 0.61 |
| 13 | 2:1 (0) | $8.0 \times 10^{-4}$ (0) | 4.0 (0) | 88.5 | 1.79 | 0.15 |
| 14 | 2:1 (0) | $8.0 \times 10^{-4}$ (0) | 4.0 (0) | 88.0 | 1.96 | 0.079 |
| 15 | 2:1 (0) | $8.0 \times 10^{-4}$ (0) | 4.0 (0) | 88.3 | 1.82 | 0.36 |

Using Design Expert software (version 10), experimental data in Table 1 were analyzed by a second-order linear polynomial regression model (Equation (7)).

$$\eta = \gamma_0 + \gamma_1 A + \gamma_2 B + \gamma_3 C + \gamma_{12} AB + \gamma_{13} AC + \gamma_{23} BC + \gamma_{11} A^2 + \gamma_{22} B^2 + \gamma_{33} C^2 \tag{7}$$

in which $\eta$ is the dependent factor (response), $\gamma_0$ is the intercept; A ($[H_2O_2]/[Fe^{2+}]$), B ($[Fe^{2+}]$) and C (pH) are the independent variables; $\gamma_1$, $\gamma_2$ and $\gamma_3$ are the coefficients of the linear part of the predicted model; $\gamma_{12}$, $\gamma_{13}$ and $\gamma_{23}$ are the interaction coefficients and $\gamma_{11}$, $\gamma_{22}$ and $\gamma_{33}$ are the quadratic coefficients. Interaction and quadratic coefficients refer to the effects of the interaction among independent variables.

Analysis of variances (ANOVAs) and significant test results for the quadratic regression equations are shown in Table 2.

**Table 2.** ANOVA of the regression model for the prediction of degree of methylene blue (MB) decolorization, rate constant of MB decolorization and *E. coli* K12 inactivation.

| Source | Sum of Squares | Df | Mean Square | *F*-Value | *p*-Value | Observations |
|---|---|---|---|---|---|---|
| **Degree of MB decolorization ($D_\%{}^{MB}$)** | | | | | | |
| Model | 627.24 | 9 | 69.69 | 975.81 | <0.0001 | significant |
| A-[$H_2O_2$]/[$Fe^{2+}$] | 397.41 | 1 | 397.41 | 5564.29 | <0.0001 | - |
| B-[$Fe^{2+}$] | 72.17 | 1 | 72.17 | 1010.43 | <0.0001 | - |
| C-pH | 69.91 | 1 | 69.91 | 978.81 | <0.0001 | - |
| AB | 6.64 | 1 | 6.64 | 92.99 | 0.0002 | - |
| AC | 0.22 | 1 | 0.22 | 3.14 | 0.1365 | - |
| BC | 3.52 | 1 | 3.52 | 49.26 | 0.0009 | - |
| $A^2$ | 58.93 | 1 | 58.93 | 825.15 | <0.0001 | - |
| $B^2$ | 0.055 | 1 | 0.055 | 0.76 | 0.4223 | - |
| $C^2$ | 13.62 | 1 | 13.62 | 190.72 | <0.0001 | - |
| Residual | 0.36 | 5 | 0.071 | - | - | - |
| Lack of Fit | 0.21 | 3 | 0.069 | 0.92 | 0.5584 | not significant |
| Pure Error | 0.15 | 2 | 0.075 | - | - | - |
| Cor Total | 627.60 | 14 | - | - | - | - |
| **Rate constant of MB decolorization ($k_{app}{}^{MB}$)** | | | | | | |
| Model | 1.17 | 9 | 0.13 | 6.47 | 0.0267 | significant |
| A-[$H_2O_2$]/[$Fe^{2+}$] | 0.019 | 1 | 0.019 | 0.94 | 0.3768 | - |
| B-[$Fe^{2+}$] | 0.37 | 1 | 0.37 | 18.64 | 0.0076 | - |
| C-pH | $8.98 \times 10^{-3}$ | 1 | $8.98 \times 10^{-3}$ | 0.45 | 0.5328 | - |
| AB | 0.20 | 1 | 0.20 | 10.09 | 0.0246 | - |
| AC | 0.024 | 1 | 0.024 | 1.18 | 0.3275 | - |
| BC | 0.089 | 1 | 0.089 | 4.45 | 0.0887 | - |
| $A^2$ | 0.40 | 1 | 0.40 | 19.76 | 0.0067 | - |
| $B^2$ | 0.025 | 1 | 0.025 | 1.25 | 0.3141 | - |
| $C^2$ | 0.069 | 1 | 0.069 | 3.43 | 0.1232 | - |
| Residual | 0.10 | 5 | 0.020 | - | - | - |
| Lack of Fit | 0.083 | 3 | 0.028 | 3.27 | 0.2430 | not significant |
| Pure Error | 0.017 | 2 | $8.48 \times 10^{-3}$ | - | - | - |
| Cor Total | 1.27 | 14 | - | - | - | - |
| **E. coli K12 inactivation in uLog units ($I_{uLog}{}^{EC}$)** | | | | | | |
| Model | 0.80 | 9 | 0.089 | 62.10 | 0.0001 | significant |
| A-[$H_2O_2$]/[$Fe^{2+}$] | $2.03 \times 10^{-3}$ | 1 | $2.03 \times 10^{-3}$ | 1.42 | 0.2870 | - |
| B-[$Fe^{2+}$] | 0.082 | 1 | 0.082 | 57.42 | 0.0006 | - |
| C-pH | 0.17 | 1 | 0.17 | 119.79 | 0.0001 | - |
| AB | 0.29 | 1 | 0.29 | 205.03 | <0.0001 | - |
| AC | $3.57 \times 10^{-3}$ | 1 | $3.57 \times 10^{-3}$ | 2.50 | 0.1745 | - |
| BC | 0.056 | 1 | 0.056 | 39.45 | 0.0015 | - |
| $A^2$ | 0.011 | 1 | 0.011 | 7.53 | 0.0406 | - |
| $B^2$ | 0.16 | 1 | 0.16 | 114.39 | 0.0001 | - |
| $C^2$ | 0.033 | 1 | 0.033 | 23.33 | 0.0048 | - |
| Residual | $7.14 \times 10^{-3}$ | 5 | $1.43 \times 10^{-3}$ | - | - | - |
| Lack of Fit | $6.78 \times 10^{-3}$ | 3 | $2.26 \times 10^{-3}$ | 12.54 | 0.0747 | not significant |
| Pure Error | $3.60 \times 10^{-4}$ | 2 | $1.80 \times 10^{-4}$ | - | - | - |
| Cor Total | 0.81 | 14 | - | - | - | - |

Df: degrees of freedom. Parameter "A" represents the [$H_2O_2$]/[$Fe^{2+}$], "B" represents the [$Fe^{2+}$] and "C" represent the pH. AC, AC, BC, $A^2$, $B^2$ and $C^2$ represent the interactions of A, B and C parameters on the responses.

Table 2 listed the results of variance analysis for the MB decolorization and *E. coli* K12 removal using the Fenton process. The values of the sum of squares demonstrate the contribution of independent

variables on responses [30]. The mean squares, which are the sums of squares divided by the degree of freedom. Adequacy of the model parameters in the present study for response variables ($D_\%^{MB}$, $k_{app}^{MB}$ and $I_{uLog}^{EC}$) was determined by the Fisher value (*F*-value), obtained by dividing the mean squares of each effect by the mean squares of error [31]. The probability critical level (*p*-value) of 0.05 was considered to reflect the statistical significance of the parameters of the proposed model. The F-values > 0.001 (975.81, 6.47 and 62.10) and *p*-values < 0.05 obtained for $D_\%^{MB}$, $k_{app}^{MB}$ and $I_{uLog}^{EC}$ responses confirming the qualification of the model to predict the decolorization of MB ($D_\%^{MB}$ and $k_{app}^{MB}$) and the inactivation of *E. coli* K12 ($I_{uLog}^{EC}$) by the Fenton reaction. In addition, the validity of the model is confirmed by the *p*-value of the lack of fit with values greater than the lowest limit of fit as recommended (>0.05) [32]. As a result, the models developed in this work for predicting the $D_\%^{MB}$, $k_{app}^{MB}$ and $I_{uLog}^{EC}$ by the Fenton process were considered adequate. These models can be described as shown in Table 3 with coded three factors.

**Table 3.** Statistical results of the proposed models in terms of the coded factors.

| Response | Proposed Quadratic Model | $R^2$ | $R_{adj}^2$ |
|---|---|---|---|
| $D_\%^{MB}$ | $88.3 + 7.05A + 3.00B - 2.96C - 1.29AB + 0.24AC - 0.94BC - 4.00A^2 - 0.12B^2 + 1.92C^2$ | 0.9994 | 0.9984 |
| $k_{app}^{MB}$ | $1.86 + 0.048A + 0.22B + 0.034C - 0.22AB - 0.077AC - 0.15BC - 0.33A^2 - 0.082B^2 - 0.14C^2$ | 0.9210 | 0.7787 |
| $I_{uLog}^{EC}$ | $0.20 - 0.061A + 0.030B + 0.10C - 0.24AB - 0.003AC + 0.11BC + 0.22A^2 + 0.15B^2 - 0.075C^2$ | 0.9911 | 0.9752 |

The ANOVA results of three parameters ($D_\%^{MB}$, $k_{app}^{MB}$ and $I_{uLog}^{EC}$) showed that the significant ($p < 0.05$) response surface models with high $R^2$ value (0.9210–0.9994) were obtained as shown in Table 3, ensuring a satisfactory adjustment of the quadratic models to the experimental data. The $R_{adj}^2$ values (0.7787–0.9984) obtained suggests that the three proposed models had an adequate predictive capacity. Even more, plots comparing the experimental and predicted values for $D_\%^{MB}$, $k_{app}^{MB}$ and $I_{uLog}^{EC}$ indicated a good agreement between experimental and predicted data from the model (Figure 1). Therefore, this finding indicates high correlation and adequacy of the proposed model to predict performance of the Fenton process ($D_\%^{MB}$, $k_{app}^{MB}$ and $I_{uLog}^{EC}$).

## 2.2. Effect of Variables on MB Decolorization ($D_\%^{MB}$)

Figure 2a showed the standardized effects of the components and their contribution to the $D_\%^{MB}$ in a Pareto chart. The sign of standardized effects in Pareto chart, + (favorable effect) or − (unfavorable effect), along the length of the bars provided the physical meaning of model terms. In this Pareto chart, we saw that A, B, C, AB, BC, AA and CC crossed the reference line ($p = 0.05$). It is evident that the most important model term was A ($[H_2O_2]/[Fe^{2+}]$), followed by linear terms B and C ($[Fe^{2+}]$ and pH respectively), quadratic terms corresponding to AA, etc. Thus, e.g., it can be concluded that larger A value, i.e., higher $[H_2O_2]/[Fe^{2+}]$ values, would result in an increase in the $D_\%^{MB}$.

The perturbation plots (Figure 2b) illustrates the effect of all parameters on the $D_\%^{MB}$. The positive effect means that if the effect factor level increases then the response value increases. On the other side, the negative effect means that if the effect factor level increases then the response value decreases. In other words, steep slope or curvature in a factor shows that the response is sensitive to that variable, while a relatively flat line indicates a low sensitivity of response to change with that particular variable. It was observed that the $[H_2O_2]/[Fe^{2+}]$ (A) and $[Fe^{2+}]$ values (B) had significant positive effects on $D_\%^{MB}$, while the initial pH (C) had a negative effect on this response. Gulkaya, Surucu and Dilek [29] demonstrated that $[H_2O_2/Fe^{2+}]$ is a critical parameter for improving the Fenton technology as a treatment of a carpet dyeing wastewater. Otherwise, Babuponnusami and Muthukumar [33] and Chen, et al. [34] demonstrated the positive effect of $[Fe^{2+}]$ on the degradation of phenol and Acridine Orange by Fenton technologies. In both publications it was established that an increase in the $[Fe^{2+}]$ values leads to an increase in the percentage of degradation of phenol and Acridine Orange by the Fenton reaction, in line with what has been demonstrated in the present investigation. Regards to pH in a large part of the experiments carried out by Fenton technologies exhibit an optimal pH close to

3 [24,33,34]. In these research, at pH less or greater than 3 the efficiency of Fenton technology decreases, as observed in this publication.

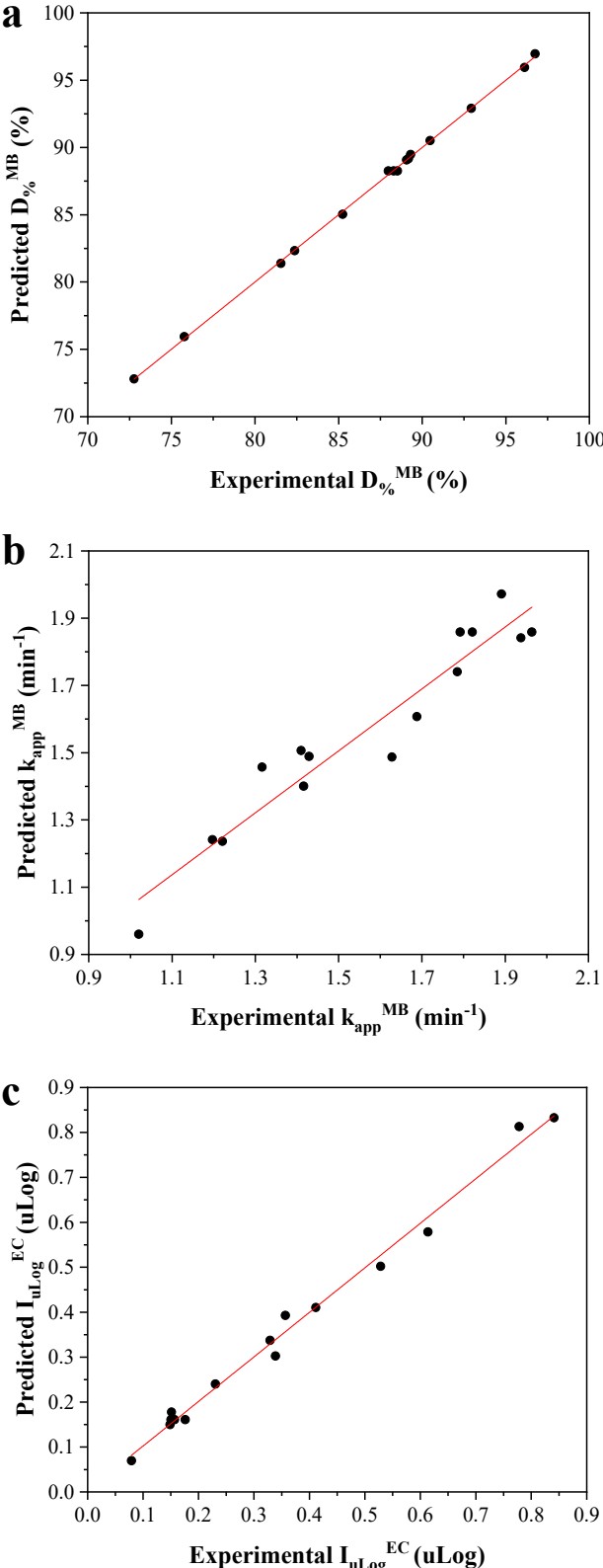

**Figure 1.** Correlations between the experimental and predicted values of (**a**) $D_{\%}^{MB}$ values, (**b**) $k_{app}^{MB}$ values and (**c**) $I_{uLog}^{EC}$ values.

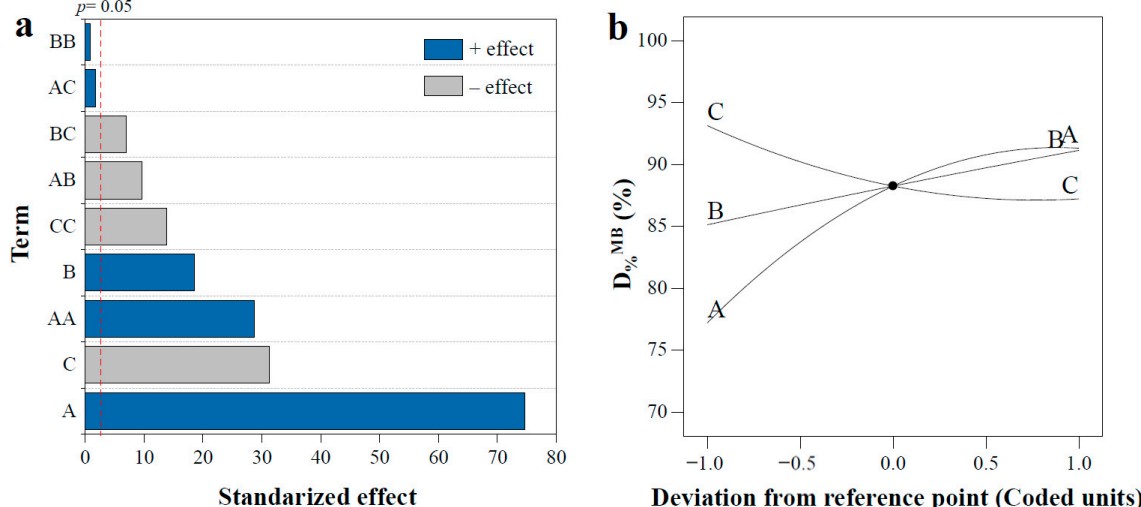

**Figure 2.** (**a**) Pareto chart showing the standardized effects of variables (first order, quadratic and interaction terms) on $D_\%^{MB}$ (vertical line represents the 95% confidence interval), and (**b**) Perturbation graphs for $D_\%^{MB}$ (A-$[H_2O_2]/[Fe^{2+}]$, B-$[Fe^{2+}]$, C-pH).

The 3D surface and contour plots in Figure S1 show the individual effects of the process variables and their interactions on the $D_\%^{MB}$. Optimum conditions determination of different variables is the main objective of the response surface methodology (RSM) study, which can affect the $D_\%^{MB}$. By considering the predicted response, $[H_2O_2]/[Fe^{2+}] = 2.9$, $[Fe^{2+}] = 1.0 \times 10^{-3}$ mol/L and pH = 3.2 of Fenton process were the optimum condition for $D_\%^{MB}$ (94.57%).

### 2.3. Effect of Variables on the MB Decolorization Rate Constant ($k_{app}^{MB}$)

Figure 3a shows the standardized effects of the components and their contribution to the $k_{app}^{MB}$ in a Pareto chart. In this Pareto chart, we saw that B, AA and AB crossed the reference line ($p = 0.05$). It is evident that the most important model terms are AA and B, followed by interaction term AB ($[H_2O_2]/[Fe^{2+}]$ and $[Fe^{2+}]$ interaction). Thus, e.g., the AA term implies that $k_{app}^{MB}$ were not influenced by $[H_2O_2]/[Fe^{2+}]$ in a linear level, but strongly influenced by this parameter in a quadratic level, i.e., a slight variation in $[H_2O_2]/[Fe^{2+}]$ will result in an increase in the $k_{app}^{MB}$.

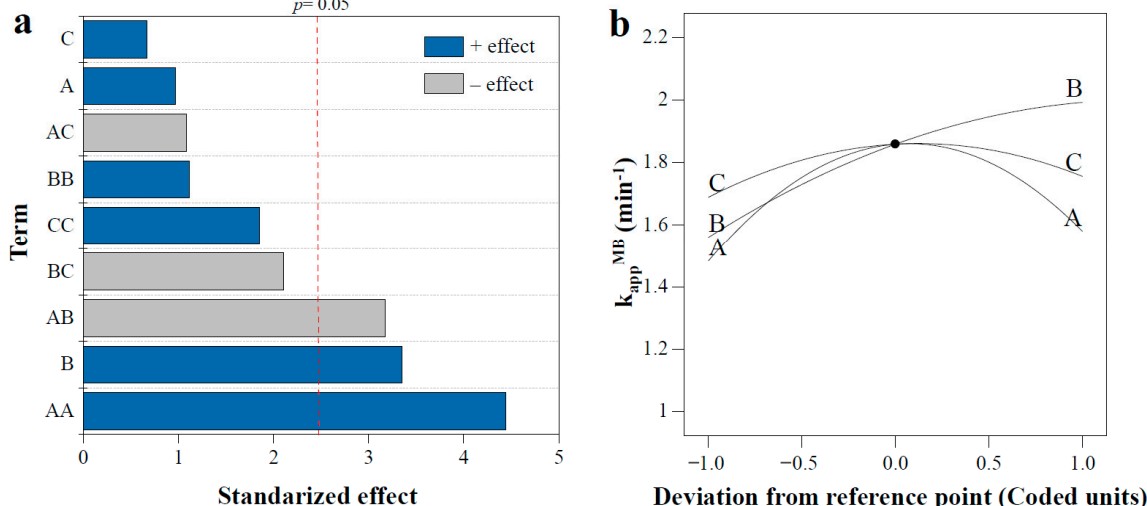

**Figure 3.** (**a**) Pareto chart showing the standardized effects of variables (first order, quadratic and interaction terms) on $k_{app}^{MB}$ (vertical line represents the 95% confidence interval), and (**b**) Perturbation graphs for $k_{app}^{MB}$ (A-$[H_2O_2]/[Fe^{2+}]$, B-$[Fe^{2+}]$, C-pH).

Figure 3b shows the perturbation plot of the effect of the parameters on the $k_{app}^{MB}$. It was observed that the $[Fe^{2+}]$ values (B) had a significant positive effect on $k_{app}^{MB}$, while the $[H_2O_2]/[Fe^{2+}]$ (B) and pH (C) had an insignificant effect on the response. It has also been reported the main role of $[Fe^{2+}]$ on the rate constants of discoloration of other dyes. For example, Tunç, et al. [35] indicated that the rate constant of acid orange 8 decolorization increased almost 10 times (0.0027–0.0267 min$^{-1}$) if $[Fe^{2+}]$ values change from $5.0 \times 10^{-6}$ to $2.5 \times 10^{-5}$ mol/L. In the same publication the rate constant of acid red 44 decolorization increased almost 4 times (0.0085–0.0331 min$^{-1}$) if $[Fe^{2+}]$ values incremented from $2.5 \times 10^{-6}$ to $2.5 \times 10^{-5}$ mol/L. Melgoza, et al. [36] reported also that the decolorization rate constant of MB increased 2.5 times (0.0014–0.0035 min$^{-1}$) if $[Fe^{2+}]$ values change from $1.0 \times 10^{-3}$ to $2.0 \times 10^{-3}$ mol/L.

Figure S2 show the individual effects of the process variables and their interactions on the $k_{app}^{MB}$ in the 3D surface and contour plots. By considering the predicted response, $[H_2O_2]/[Fe^{2+}] = 1.7$, $[Fe^{2+}] = 1.0 \times 10^{-3}$ mol/L and pH = 3.7 of the Fenton process were the optimum condition providing $k_{app}^{MB}$ (2.08 min$^{-1}$).

## 2.4. Effect of Variables on E. coli K12 Removal ($I_{uLog}^{EC}$)

Figure 4a showed the standardized effects of the components and their contribution to the $I_{uLog}^{EC}$ in a Pareto chart. In this Pareto chart, we saw that B, C, AA, BB, CC, AB and BC crossed the reference line ($p = 0.05$). It is evident that the most important model terms was AB, followed by linear term C (pH) and quadratic term BB ($[Fe^{2+}]^2$). Thus, e.g., it can be concluded that smaller C value, i.e., lower pH values, would result in an increase in the $I_{uLog}^{EC}$.

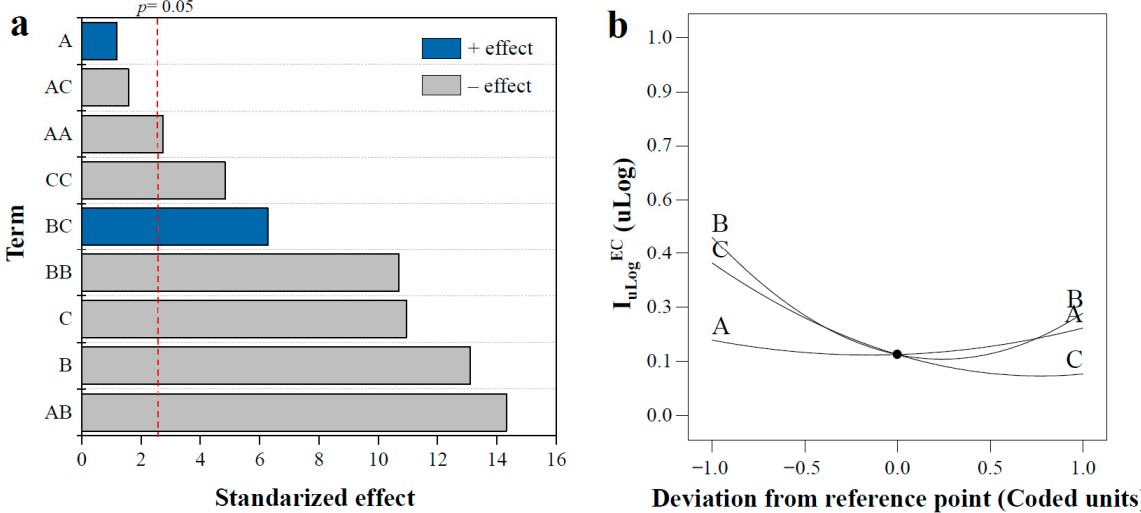

**Figure 4.** (**a**) Pareto chart showing the standardized effects of variables (first order, quadratic and interaction terms) on $I_{uLog}^{EC}$ (vertical line represents the 95% confidence interval), and (**b**) Perturbation graphs for $I_{uLog}^{EC}$ (A-$[H_2O_2]/[Fe^{2+}]$, B-$[Fe^{2+}]$, C-pH).

The perturbation plots (Figure 4b) illustrates the effect of all parameters on the $I_{uLog}^{EC}$. It was observed that the $[Fe^{2+}]$ (B) and pH (A) values had a significant negative effects on $I_{uLog}^{EC}$, while the $[H_2O_2]/[Fe^{2+}]$ (C) values did not have a statistically significant effect on this response. Asad, et al. [37] also reported that the inactivation of *E. coli* was mostly affected by Fenton technologies if the $[Fe^{2+}]$ was low. This is evident when considering Equation (5), since at high $[Fe^{2+}]$ values the activity of the ·OH formed could be inhibited.

On the other hand, it is known that the ·OH production by Fenton technology is benefited at acidic pH close to 3. Some examples of Fenton technologies applied to the inactivation of a few bacteria are presented in Table 4. Although the experiments do not have the same conditions of $[H_2O_2]/[Fe^{2+}]$

and reaction time, it is possible to identify that at lower pH values and at lower values of $[Fe^{2+}]$ the bacterial elimination efficiencies tend to increase.

**Table 4.** Examples of Fenton technologies applied to the bacteria inactivation.

| $[Fe^{2+}]$ (mol/L) | $[H_2O_2]/[Fe^{2+}]$ | pH | Time (min) | Bacteria | Reduction (uLog) | Ref. |
|---|---|---|---|---|---|---|
| $7.8 \times 10^{-4}$ | 62.2 | 3.0 | 300 | *E. coli* | 2.12 | Blanco, et al. [38] |
| $7.2 \times 10^{-5}$ | 20.4 | 3.5 | 100 | *E. faecalis* | 1.0 | Ortega-Gómez, et al. [39] |
| $1.0 \times 10^{-3}$ | 20 | 7.0 | 25 | *E. coli* | 0.108 | Ahmad and Iranzo [40] |
| $5.0 \times 10^{-3}$ | 10 | 8.5 | 1440 | *E. coli* | 0.36 | Cengiz, et al. [41] |

Figure S3 show the individual effects of the process variables and their interactions on the $I_{uLog}^{EC}$ in the 3D surface and contour plots. By considering the predicted response, $[H_2O_2]/[Fe^{2+}] = 2.9$, $[Fe^{2+}] = 7.6 \times 10^{-4}$ mol/L and pH = 3.2 of the Fenton process were the optimum condition providing $I_{uLog}^{EC}$ (0.89 uLog).

## 2.5. Analysis of Optimization and Model Validation

The optimal conditions obtained for MB decolorization and *E. coli* K12 inactivation are different for each of the responses studied. These results indicate that although some authors have suggested that it is possible to analyze the bacteria inactivation of AOP by extrapolating from dye decolorization [42], these processes have differences. The results in the present study (Table 1) show that if for example experiments #1 ($[H_2O_2]/[Fe^{2+}] = 1:1$, $[Fe^{2+}] = 6.0 \times 10^{-4}$ mol/L and pH = 4.0) and #10 ($[H_2O_2]/[Fe^{2+}] = 2:1$, $[Fe^{2+}] = 1.0 \times 10^{-3}$ mol/L and pH = 3.0) are compared, the MB decolorization reaches 72.8% and 96.7% respectively, while in the same experiments the *E. coli* K12 inactivation reaches 61.3% (0.412 uLog) and 33.4% (0.176 uLog), i.e., when the MB decolorization is high the *E. coli* K12 inactivation tends to be low, and vice versa. Similar compartments can be observed when comparing (Table 1), for example, experiments #3 ($[H_2O_2]/[Fe^{2+}] = 1:1$, $[Fe^{2+}] = 1.0 \times 10^{-3}$ mol/L and pH = 4.0) and #14 ($[H_2O_2]/[Fe^{2+}] = 2:1$, $[Fe^{2+}] = 8.0 \times 10^{-4}$ mol/L and pH = 4.0). These observations support what has been observed in other studies on the complexity involved in the bacteria inactivation compared to the recalcitrant compounds elimination [43].

To validate the model obtained by the Box–Behnken optimization technique, experiments were carried out with the suggested optimum values of independent variables. Table 5 shows the optimal conditions predicted by the models, the predicted response value and the response value obtained experimentally (Table 5).

**Table 5.** Results of validation experiments under optimized conditions.

| Conditions | $[H_2O_2]/[Fe^{2+}]$ | $[Fe^{2+}]$ (mol/L) | pH | Experimental Value | Predicted Value | Error |
|---|---|---|---|---|---|---|
| $D_\%^{MB}$ | 2.9 | $1.0 \times 10^{-3}$ | 3.2 | 94.57 | 97.12 | −2.55 |
| $k_{app}^{MB}$ | 1.7 | $1.0 \times 10^{-3}$ | 3.7 | 2.08 | 2.02 | 0.06 |
| $I_{uLog}^{EC}$ | 2.9 | $7.6 \times 10^{-4}$ | 3.2 | 0.89 | 0.92 | −0.03 |

The result obtained from experiments for all response parameters was in agreement with the model prediction. Low errors showed the model and parameters could accurately reflect on the three responses analyzed.

## 2.6. Effect of $[H_2O_2]/[Fe^{2+}]$ on Responses

The increasing of $[H_2O_2]/[Fe^{2+}]$ value (studied at molar ratio 1:1, 2:1 and 3:1) showed a significant positive effect only on the values of $D_\%^{MB}$. Gulkaya, Surucu and Dilek [29] studied the treatment of wastewater from carpet dyes by Fenton technologies, also finding that $[H_2O_2]/[Fe^{2+}]$ plays a crucial role in the removal of dyes. Considering the above, it is possible to indicate that it is necessary to maintain a considerable concentration of $H_2O_2$, to ensure that the ·OH production is maintained for a longer

time, obtaining a high efficiency in the discoloration of MB. However, the values of $[H_2O_2]/[Fe^{2+}]$ do not seem to significantly affect the $k_{app}^{MB}$ and $I_{uLog}^{EC}$ values.

## 2.7. Effect of [Fe²⁺] on Responses

Statistical analyses identified $[Fe^{2+}]$ as an important factor for the three types of responses studied $(D_{\%}^{MB}, k_{app}^{MB}$ and $I_{uLog}^{EC})$. For both the $D_{\%}^{MB}$ analysis and for $k_{app}^{MB}$ the increasing of $[Fe^{2+}]$ value (studied at $6.0 \times 10^{-4}$, $8.0 \times 10^{-4}$ and $1.0 \times 10^{-3}$ mol/L) exhibited a significant positive effect on these responses, while for the $I_{uLog}^{EC}$ analysis this parameter showed a significant negative effect. In Equation (1) it is observed that the formation of ·OH due to the oxidation of $Fe^{2+}$ to $Fe^{3+}$ in the presence of $H_2O_2$ will increase with a higher concentration of $Fe^{2+}$. A single ·OH attack on the MB structure leads to its decolorization ($D_{\%}^{MB}$ and $k_{app}^{MB}$), which seems to agree with the importance of $[Fe^{2+}]$ found in the present research. However, if the ·OH also had a preponderant role in the inactivation of *E. coli* K12, then the $[Fe^{2+}]$ values should not have a significant negative effect on the $I_{uLog}^{EC}$ response. Some authors [44–47] suggest that although ·OH oxidize most simple organic compounds (such as recalcitrant compounds), the inactivation of a bacterium (such as *E. coli* K12) is not directly affected by the production of these radicals. These researchers indicate that the formation of singlet oxygen ($^1O_2$) is responsible for the inactivation of bacteria. Consequently, the $I_{uLog}^{EC}$ decreases with increasing values of $[Fe^{2+}]$, since excess $Fe^{2+}$ could react with the ·OH formed (Equation (5)), decreasing the possibility that the ·OH react to form $^1O_2$ (Equations (3) and (4)).

## 2.8. Effect of pH on Responses

The pH value (studied at pH 3.0, 4.0 and 5.0) showed a significant effect on the $D_{\%}^{MB}$ and $I_{uLog}^{EC}$ responses but did not show a significant effect on the $k_{app}^{MB}$ response. The Fenton reaction ($Fe^{2+}$ and $H_2O_2$), with a rate constant 76 L·mol$^{-1}$s$^{-1}$ [48], form ·OH quickly, consume $Fe^{2+}$ and produce $Fe^{3+}$. The Fenton-like reaction ($Fe^{3+}$ and $H_2O_2$) has a much slower rate constant (0.01 L·mol$^{-1}$s$^{-1}$) than the Fenton reaction [48] and it only produces $O_2\cdot^-$, a much less reactive radical than ·OH. Additionally it has been established that the species of Fe(II) that prevails in the working pH range (3.0–5.0) is $Fe^{2+}$ [49], while in the same pH range the speciation of Fe(III) demonstrates the formation of $Fe(OH)^{2+}$ and $Fe(OH)_2^+$ species, species that are less reactive than $Fe^{3+}$ [22]. Based on this information, it is expected that the rate of ·OH formation from the Fenton reaction, at least in the first minutes of reaction that directly influence the determination of $k_{app}^{MB}$, will not be greatly altered when changing system pH between 3.0 and 5.0. However, $D_{\%}^{MB}$ and $I_{uLog}^{EC}$, which are obtained in a final time of 15 min, will be influenced by both the Fenton reaction and the subsequent Fenton-like reaction. Considering this, the participation of the Fenton-like reaction implies that the pH and its effect on Fe(III) speciation have a greater influence on the $D_{\%}^{MB}$ and $I_{uLog}^{EC}$ responses, as observed in this investigation.

## 3. Materials and Methods

### 3.1. Chemicals and Materials

Iron sulfate heptahydrate ($FeSO_4\cdot7H_2O$), hydrogen peroxide ($H_2O_2$, 30%), sodium hydroxide (NaOH), hydrochloric acid (HCl), Luria Bertani (LB) medium and methylene blue ($C_{16}H_{18}N_3SCl\cdot3H_2O$) were purchased from Merck S.A. (Santiago, Chile).

### 3.2. Fenton Experiments

Methylene blue, a dye that does not generate toxic byproducts when reacting with ·OH [50–52], was used as a model of recalcitrant compound, while *E. coli* K12, a non-pathogenic *E. coli* [53], was used as a model of bacteria. Experiments were performed in 20 mL glass reactors containing the MB solution ($5.0 \times 10^{-5}$ mol/L) or *E. coli* K12 ($10^6$ CFU), kept under magnetic stirring at room temperature (25 °C) [43,44]. First, $FeSO_4\cdot7H_2O$ solution was added to each sample according to the experimental design. The pH of each sample was adjusted by using NaOH (0.25 mol/L) or HCl (0.10 mol/L)

solutions. Reactions were started by adding an aliquot of $H_2O_2$ solution. After the experimental time elapsed (15 min), for *E. coli* K12 analysis, 0.2 mL of each sample was collected for its enumeration. The decolorization of MB was studied by determining its kinetic constants of color decay and the degree of decolorization. After 2 min of maintaining the reaction under constant agitation, samples (3.0 mL) were withdrawn, and immediately injected into a cuvette for analysis at time intervals of 3, 6, 9, 12 and 15 min. The analyses in samples were performed spectrophotometrically by UV–Vis spectrophotometry (Shimadzu UV-1800, Shimadzu Inc., Kyoto, Japan) at 668 nm using quartz cells with path lengths of 1 cm. A calibration curve was constructed ($5.30 \times 10^{-7}$–$1.30 \times 10^{-5}$ mol/L; $R^2 = 0.999$). Fitting decolorization kinetics and the rate constant was obtained by Sigma Plot 11.0 software (Systat Software, Inc., San Jose, CA, USA).

### 3.3. Detection and Enumeration of E. coli K12

Strain samples were stored in cryo-vials containing 20% glycerol at −20 °C. To prepare the bacterial pellet for the experiments, one colony was picked from the precultures and loop-inoculated into a 50 mL sterile PE eppendorf flask containing the Luria Bertani (LB) medium. The flask was then incubated aerobically at 37 °C and 150 rpm in a shaker incubator (Gerhardt THO500, Gerhardt GmbH & Co., Königswinter, Germany) until the stationary physiological phase was reached. After 24 h, cells were centrifuged (SIGMA 2-16P, Sigma Laborzentrifugen GmbH, Steinheim, Germany) and diluted until optical density 0.5 a.u. (i.e., $10^6$ CFU/mL) at 600 nm [43,44]. Component of LB medium included sodium chloride (10 g), tryptone (10 g) and yeast extract (5 g) in 1 L of deionized water; this solution was then sterilized by autoclaving for 20 min at 121 °C. The bacterial pellet was resuspended and washed three times with a saline solution (NaCl/KCl). The final pellet was resuspended in saline solution. This procedure resulted in a cell density of approximately $10^9$ colony forming units (CFU) per milliliter. The pH of the solution was adjusted to 7.0 and the solution was then sterilized by autoclaving for 30 min at 121 °C. The bacterial solution was diluted in reactors to the required cell density corresponding to $10^6$ CFU/mL [43,44].

CFU were performed by plating on plates (PCA method). Of the samples 0.2 mL was withdrawn. Samples were diluted (10% *v/v*) and 0.1 mL poured on plates. Plates were aerobic incubated for 24 h at 37 °C (Heraeus B6, Kendro, Langenselbold, Germany) and the CFU were counted manually. All experiments were performed in triplicates. The enumeration of colonies was expressed as CFU (colony forming units) per 100 mL of sample. These concentrations were transformed to $\log_{10}$ and the removal of bacteria, uLog = $\log(N_t/N_0)$, was calculated from the initial bacteria concentration ($N_0$) and the remaining bacteria population at "t" time ($N_t$).

### 3.4. Experimental Design

To determine the optimal experimental conditions for the decolorization of MB and the inactivation of *E. coli* K12 by Fenton technology, a Box–Behnken design was performed. pH, $Fe^{2+}$ concentration ($[Fe^{2+}]$, mol/L) and molar concentration ratio of $Fe^{2+}$ and $H_2O_2$ ($[H_2O_2]/[Fe^{2+}]$) were selected as independent variables in the experimental design (Table 6).

**Table 6.** Independent variables and levels used in the Box–Behnken design for Fenton technology.

| Variable | Coded | \multicolumn Coded Factor Level | | |
|---|---|---|---|---|
| | | −1 | 0 | 1 |
| $[H_2O_2]/[Fe^{2+}]$ (A) | $X_1$ | 1:1 | 2:1 | 3:1 |
| $[Fe^{2+}]$ (mol/L) (B) | $X_2$ | $6.0 \times 10^{-4}$ | $8.0 \times 10^{-4}$ | $1.0 \times 10^{-3}$ |
| pH (C) | $X_3$ | 3.0 | 4.0 | 5.0 |

Three replicates were performed at the central point, with 15 runs performed for each study. The chosen levels of the independent variables were based on literature reports [54]. The experimental

responses were the degree of MB decolorization ($D_\%^{MB}$), the apparent kinetic constant of MB decolorization ($k_{app}^{MB}$), and removal of bacteria in uLog units ($I_{uLog}^{EC}$) for variables showed in Table 6.

A second-order linear polynomial regression model (Equation (7)) was obtained to analyze the data. Data were statistically evaluated and an analysis of variance (ANOVA) was applied at with a confidence level of 95% using software Design Expert version 10 (Stat-Ease Inc., Minneapolis, MN, USA). Responses of the experimental tests were compared to the estimated values, and the fit of model was assessed. Experimental tests, performed under optimal conditions, were performed to achieve maximal $D_\%^{MB}$, $k_{app}^{MB}$ and $I_{uLog}^{EC}$.

## 4. Conclusions

The present study provided a comprehensive description regarding the application of the Fenton technology as a process for MB decolorization and *E. coli* K12 inactivation in aqueous solutions at different $[H_2O_2]/[Fe^{2+}]$ values (1.0, 2.0 and 3.0), $[Fe^{2+}]$ values ($6.0 \times 10^{-4}$, $8.0 \times 10^{-4}$ and $1.0 \times 10^{-3}$ mol/L) and pH values (3.0, 4.0 and 5.0) up to 15 min of reaction. It was found that the Box–Behnken model could effectively predict and optimize the performance of Fenton technology for MB decolorization and *E. coli* K12 inactivation.

The maximum $D_\%^{MB}$ of 94.57% was predicted at $[H_2O_2]/[Fe^{2+}] = 2.9$, $[Fe^{2+}] = 1.0 \times 10^{-3}$ mol/L and pH = 3.2; for $k_{app}^{MB}$ the maximum of 2.08 min$^{-1}$ was predicted at $[H_2O_2]/[Fe^{2+}] = 1.7$, $[Fe^{2+}] = 1.0 \times 10^{-3}$ mol/L and pH = 3.7 and the maximum $I_{uLog}^{EC}$ of 0.89 uLog was predicted at $[H_2O_2]/[Fe^{2+}] = 2.9$, $[Fe^{2+}] = 7.6 \times 10^{-4}$ mol/L and pH = 3.2. This analysis revealed good agreement between experimental results and the RSM predictions, further illustrating that RSM is a suitable approach to optimize the MB decolorization and *E. coli* K12 inactivation.

The Pareto and perturbation analysis of the model terms showed that all parameters analyzed have different effects on the responses. The $[H_2O_2]/[Fe^{2+}]$ values show a significant positive effect only on $D_\%^{MB}$. The pH values show a significant negative effect on $D_\%^{MB}$ and $I_{uLog}^{EC}$, which could involve the main role of speciation of Fe(II) and Fe(III) species in the total process of MB decolorization and *E. coli* K12 inactivation by Fenton technology. The positive and negative significant effect of the $[Fe^{2+}]$ values on the MB decolorization ($D_\%^{MB}$ and $k_{app}^{MB}$) and *E. coli* K12 inactivation ($I_{uLog}^{EC}$) respectively, suggest that different oxidizing species are involved in these processes.

Thus, considering that bacteria are larger than dye molecules, the complex self-repair mechanisms of bacteria and the different external structures of bacteria compared to the dyes structure, the *E. coli* inactivation proved to be less effective than MB decolorization by Fenton processes.

**Supplementary Materials:** The following are available online at http://www.mdpi.com/2073-4344/10/12/1483/s1, Figure S1: Three-dimensional response surface plots (a, c and e) and their corresponding contour plots (b, d and f) representing de modeled $D_\%^{MB}$ as a function of: $[H_2O_2]/[Fe^{2+}]$ and $[Fe^{2+}]$ (a, b), pH and $[H_2O_2]/[Fe^{2+}]$ (c, d), $[Fe^{2+}]$ and pH (e, f) at central point values of other parameters, Figure S2: Three-dimensional response surface plots (a, c and e) and their corresponding contour plots (b, d and f) representing de modeled $k_{app}^{MB}$ as a function of: $[H_2O_2]/[Fe^{2+}]$ and $[Fe^{2+}]$ (a, b), pH and $[H_2O_2]/[Fe^{2+}]$ (c, d), $[Fe^{2+}]$ and pH (e, f) at central point values of other parameters, Figure S3: Three-dimensional response surface plots (a, c and e) and their corresponding contour plots (b, d and f) representing de modeled $I_{ulog}^{EC}$ as a function of: $[H_2O_2]/[Fe^{2+}]$ and $[Fe^{2+}]$ (a, b), pH and $[H_2O_2]/[Fe^{2+}]$ (c, d), $[Fe^{2+}]$ and pH (e, f) at central point values of other parameters.

**Author Contributions:** Conceptualization, P.S. and G.V.; methodology, P.S. and G.V.; software, P.S. and J.L.F.; validation, P.S. and J.L.F.; formal analysis, P.S. and G.V.; investigation, J.L.F.; writing—original draft preparation, P.S. and G.V.; writing—review and editing, P.S. and G.V.; supervision, G.V.; project administration, G.V.; funding acquisition, P.S. and G.V. All authors have read and agreed to the published version of the manuscript.

**Funding:** This research was funded by ANID/FONDAP/15130015 and ANID FONDECYT/Postdoctorado 3180566.

**Acknowledgments:** José Luis Frontela thanks ERASMUS mobility program.

**Conflicts of Interest:** The authors declare no conflict of interest.

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
