# Peer review of "Optimization of Fenton Technology for Recalcitrant Compounds and Bacteria Inactivation"

_catalysts, doi:10.3390/catal10121483_

Round 1
Reviewer 1 Report
Dear Authors,
An interesting proposal to use Box-Behnken plans to evaluate and optimize process parameters could be taken into consideration. However, these plans are very economical and therefore particularly useful when measurements are expensive and the number should be limited to what is necessary. With regard to the experiment described in the paper (which are not expensive), I do not find any indication for the use of Box-Behnken plans.
In terms of novelty in the field of the experiment, the paper does not bring any new knowledge. It is commonly believed that the use of Fenton's reagent is an effective method of degrading most pollutants and competitive to other methods of enhanced oxidation, due to the general availability of reagents, simplicity and no need for specialized equipment. The use of Fenton technology for discoloration of aqueous solutions (wastewater) and inactivation of bacteria has been the subject of research by many scientists and has been described in many papers. Meanwhile, the described experiment was based only on model solutions, was carried out on a small laboratory scale, so the research should be considered basic, while the technology is already much further. The use of Fenton's reagent works well in the treatment of colored, as well as colored and highly turbid wastewater. It was possible to carry out research on real samples, e.g. sewage, in which many complex pollutants are contained, not only colored compounds
In the experiment, the authors took commonly used reagents - H2O2 and FeSO4 · 7H2O. If, for example, they looked at least for modification of Fenton's reagent with the use of alternative sources of H2O2 and Fe (which could make this method easier to apply and more effective) and demonstrated effectiveness in the treatment of e.g. colored wastewater, it would be a novelty.
In the Introduction section, little attention was paid to the literature review of the research on the application of Fenton technology.
Detailed remarks:
- The authors did not explain why they adopted a solution with a concentration of 0·10-5 mol/L as the MB model, and a solution of E. coli K12 with a cell density of 106 CFU/ml as a bacteria model.
- The authors did not explain why they adopted the test range of pH 3-5 only.
- The authors could at least verify the results of the experiment carried out in model solutions on a real sample, e.g. industrial sewage.
- The dependence of the pollutant degradation effect on temperature was not taken into account.
- The possibility of the formation of more toxic oxidation intermediates or the possibility of the formation of toxic by-products was not addressed.
- Why was the residual H202 content not controlled after the oxidation process?
In its current form, the article does not meet the scientific requirements for publication.
Best regards,
Author Response
Dear Reviewer 1,
Attached you will find the answers to the questions. Thank you very much for all and help with the publication.
With my best regards,
Dr. Gladys Vidal

Reviewer 2 Report
The manuscript «Optimization of Fenton technology for recalcitrant compounds and bacteria inactivation» by Pablo Salgado, José Luis Frontela and Gladys Vidal is devoted to the optimization of the Fenton process for methylene blue and E. coli K12 removal from water. The purification of wastewater is a global issue during the last two decades and now.
This work has great importance for the improvement the ecological situation, in particular the removal of organic pollutants and bacteria from the waste water. The advantage of this work is the application of mathematical methods. The authors created mathematical models, which describe the removal of organic pollutants and bacteria, depending on the oxidation process conditions. The material is a good example of combination of chemistry and mathematics.
Despite the relevance and undoubted advantages of the work, it cannot be accepted for publication in journal “Catalysts” in its current form.
There are some comments and remarks that should be corrected and carefully checked (see below).
I kindly ask the authors carefully look at the attached pdf file, as it contains important comments.
The presented material is redundant and contains a lot of unnecessary information, it needs to be shortened. Some of the information can be transferred to the “Supplementary materials”. For example there are lot of plots and their description, which can be omitted or transferred.
Please, give in the abstract some numerical data, for example, concentration and pH intervals used, optimized parameters and indicators.
IuLogEC - is unclear parameter. Maybe it would be better to use the “degree of E.coli removal”?
The main body text contains unsuccessful and incorrect sentences (marked by yellow in the pdf-file).
Lines 60-61 “·OH has a standard reduction potential of 2.8 V” – check this sentence and provide the data for the reference electrode.
Lines 107-108: Please, give some description of the following indicators D%MB, kappMB and IuLogEC.
Table 1: Where is the variable parameters?
Table 2: Write as a note at the end of the table designations and what parameters they mean.
Figure 2: What is the meaning of the parameters AA, AB, AC, BC etc.?
Figure 2: Is the contribution depended on the dimensions of the given parameters?
Lines 169-179: This description must be shortened. Please give only some conclusions from the 3D plots.
Lines 192-194: How in this case you can explain the effect and the meaning of the “AA” parameter?
Line 315: What is the pH interval, which was used in the work?
Lines 349-350: The “Results and Discussion” doesn’t contain any information regarding kinetic parameters of the Fenton process. At the same time you determined rate constants. Please, add information about the kinetics, as well as the difference between the kinetics of the degradation on MB and E. coli.

Author Response
Dear Reviewer 2,
Attached you will find the answers to the questions. Thank you very much for all and help with the publication.
With my best regards,
Dr. Gladys Vidal

Reviewer 3 Report
The manuscript entitled “Optimization of Fenton technology for recalcitrant compounds and bacteria inactivation” presents the use of Box-Behnken design (BBD) to evaluate and optimize the effects of three parameters important in Fenton technology: Fe2+ concentration, molar ratio between H2O2 and Fe2+ and pH. Methylene blue was used as a model of recalcitrant compound and Escherichia coli as model of bacteria, in order to show the effects of the main process variables influencing the Fenton technology. The authors found that Box-Behnken model could predict and optimize the performance of Fenton technology for methylene blue decolorization and Escherichia coli inactivation. They proved the bacteria inactivation was be less effective that dyes decolorization by Fenton technology under the tested conditions.
The subject of the manuscript is interesting, the authors present the appropriate literature related to their research topic, characterization of obtained results is sufficient.
I recommend this manuscript for publication.
Author Response
Dear Reviewer 3,
Attached you will find the answers to the questions. Thank you very much for all and help with the publication.
With my best regards,
Dr. Gladys Vidal

Round 2
Reviewer 1 Report
Dear Authors,
I apologize to the Authors for my negative opinion on the manuscript and for responding to my comments. I believe that any scientific work should show aspects of novelty, originality and mainly it should clearly indicate the direction and possibility of their implementation.
The authors responded only to specific comments, but did not comment on the key allegation, namely the aspect of novelty, originality and the fact that they go back many years when conducting model research in the presented manner. The results of such studies were published in the eighties and nineties of the last century. The authors did not move forward in their research. Nowadays, research with the use of Fenton's reagent are well advanced. They are run in more complex systems (with the presence of other impurities as well).
The Authors' answers are unfortunately not acceptable.
QUESTION 1. The authors did not explain why they adopted a solution with a concentration of 5.0·10-5 mol/L as the MB model, and a solution of E. coli K12 with a cell density of 106 CFU/ml as a bacteria model.
ANSWER 1. The concentrations of methylene blue and bacteria were selected based on other publications in which advanced oxidation processes were compared oxidizing methylene blue and eliminating E. coli bacteria. We add the number of references in the respective sentences where the concentrations of methylene blue and E. coli are mentioned: lines 339, 358, 365.
COMMENT: The authors reply, that they selected the appropriate values of the process parameters on the basis of other publications. This argument cannot be considered acceptable. The selection of concentrations should be based on the purposefulness of the research. The research is conducted so that, on the basis of the obtained results, it is possible to develop a technology that can be implemented. Therefore, researchers' choices may differ. Even on this point, there is no original contribution from the Authors.
QUESTION 2. The authors did not explain why they adopted the test range of pH 3-5 only.
ANSWER 2. In lines 172-174 it is mentioned that at pH close to 3 the Fenton reaction reaches its maximum performance. In lines 87-88 it is mentioned that the Fe2+ species is essential for the optimal performance of the Fenton reaction, while in lines 318-320 it is mentioned that the Fe2+ species prevails between pH 3 to 5.
COMMENT: The answer is not satisfactory and does not explain everything. Indeed, most often the pH range from 3 to 5 is considered optimal. However, depending on the substrate to be oxidized, the optimal pH range may in some cases assume other marginal values, i.e. pH 2 - 4, e.g.
Park T.J., Lee K.H., Jung E.J., Kim C.W.: Removal of refractory organics and color in pigment wastewater with Fenton oxidation. Wat. Sci. Technol., 39, 189 (1999).; Kuo W.G.: Decolorizing dye wastewater with Fenton’s reagent. Wat. Res., 26, 881 (1992).; Miller C.M., Valentine R.L., Roehl M.E., Alvarez P.J.J.: Chemical and microbiological assessment of pendimethalin-contaminated soil after treatment with Fenton reagent. Wat. Res., 30, 2579 (1996).; Sheu S-H., Weng H-S.: Treatment of olefin plant caustic by combination of neutralization and Fenton reaction. Wat. Res., 35, 2017 (2001).; Kwon B.G., Lee D.S., Kang N., Yoon J.: Characteristics of p-chlorophenol oxidation by Fenton’s reagent. Wat. Res., 33, 2110 (1999).
and even pH 5 - 6 e.g. Preis S., Kamenev S., Kallas J.: Oxidative purification of wastewaters containing phenolic compounds from oil shale treatment. Environ . Technol., 15, 135 (1994).
In some cases, the effectiveness of Fenton's reagent is the same over a fairly wide pH range, e.g.3 – 9 Nerud F., Baldrian P., Gabriel J., Ogbeifun D.: Decolorization of synthetic dyes by the Fenton reagent and the Cu/pyridine/H20 2 system. Chemosphere , 44, 957 (2001).
QUESTION 3. The authors could at least verify the results of the experiment carried out in model solutions on a real sample, e.g. industrial sewage.
ANSWER 3. Although it is a point that could have been addressed, the objective of this work was to study and compare how the Fenton reaction and its main parameters affected the decolorization of methylene blue and the inactivation of E. coli, since both have completely different structures. We intend to do studies on real wastewater in future investigations. Besides, due to the recess of experimental work in Chilean universities caused by the Covid-19 pandemic, it would be impossible for us to do an experiment that evaluates the performance of the Fenton reaction in real samples.
Comment: The authors did not seem to understand the suggestion. The research presented in the manuscript does not add any new knowledge. Therefore, research with the real sample would give a chance for a positive assessment of this paper.
QUESTION 4. The dependence of the pollutant degradation effect on temperature was not taken into account.
ANSWER 4. Although in a laboratory it is not difficult to control the temperature of the systems, it is difficult in systems at an industrial level (with a greater depth of our studies it is expected to be able to apply these systems at an industrial level), in addition to adding a substantial cost in the processes due to increased energy use. Additionally, increasing one more parameter to the study meant a substantial increase in the total number of experiments. For these reasons, it seemed to us that the best option was not to consider the temperature dependence in the Fenton reaction, since it is better to study parameters that are easier and more feasible to control in a possible application in real systems.
Comment: The answer is not satisfactory. Temperature is an important parameter. Its influence cannot be ignored, because the speed of Fenton's reaction increases with increasing temperature. However, after exceeding the temperature of 40-50°C, the efficiency of the reaction usually begins to decline. In practice, it is recommended to use Fenton's reagent in the range of 20-40°C. However, there are reports of much higher optimal temperatures such as 70°C, and even 90°C. Therefore, the influence of temperature should have been taken into account in the research.
QUESTION 5. The possibility of the formation of more toxic oxidation intermediates or the possibility of the formation of toxic by-products was not addressed.
ANSWER 5. We appreciate your comment, as it is a very important point to consider when studying the degradation of organic pollutants. However, it is well documented that the intermediates produced by ·OH in the degradation of methylene blue, as occurs in this investigation, do not show toxicity.
https://doi.org/10.1016/j.apcatb.2017.01.075
https://doi.org/10.1016/j.chemosphere.2018.11.098
https://doi.org/10.1016/j.jwpe.2020.101272
For these reasons, we ruled out doing a study of intermediaries and their toxicity in this research.
Comment: This is the information that should have been included in the work, so that there is no doubt.
QUESTION 6. Why was the residual H202 content not controlled after the oxidation process?
ANSWER 6. The residual H2O2 content was not controlled because we knew from other work that:
- The H2O2 concentration must be higher than the Fe2+ concentration for a good performance of the Fenton reaction (https://doi.org/10.1039/B817287K)
- The optimal H2O2-Fe2+ molar ratio depends on the pollutant to be oxidized (https://doi.org/10.1016/j.jhazmat.2006.01.006)
- It has been determined that the H2O2: Fe2+ molar ratio for optimal ·OH production in the Fenton reaction is at least 90:1 (https://www.jbc.org/content/265/23/13589.short)
- To degrade methylene blue, it has been determined that the optimal molar ratio of H2O2: Fe2+ must be at least 12.5:1 (https://doi.org/10.1016/S0304-3894(01)00202-3)
Therefore, based on the evidence we had, we assumed that in the range of the molar ratios of H2O2:Fe2+ used in this research we would not have residual H2O2. In fact, the optimal H2O2:Fe2+ molar ratios found for methylene blue degradation are close to 3, which could support our previous conclusions.
Comment: The answer confirms that the research is not the product of the authors' original concept. Moreover, scientific reliability requires the assumptions be checked and verified.
Author Response
Editors
Catalyst
Concepción (Chile), November 30th, 2020
Dear Reviewer 1,
Please find herewith the secod evised version of the manuscript titled “Optimization of Fenton technology for recalcitrant compounds and bacteria inactivation” by Pablo Salgado, José Luis Frontela and Gladys Vidal which was assigned reference no. catalysts-978824
We want to thank the reviewers for so carefully reading the manuscript and their valuable suggestions. All the comments enclosed were attended and the paper was carefully checked according to the editorial suggestions.
With best regards,
Gladys Vidal

Reviewer 2 Report
Dear Authors,
thank you for the work done to improve the quality of the manuscript. Despite of this, the article still needs major revision.
The “Results and discussion” section is redundant and difficult to perceive and understand. I would recommend to make it shorter.
I think that it would be better to combine the data from Tables 1 and 7 in one Tables.
Casting of the second decimal place in pH values is not correct in your case. Also please check the accuracy of the other measured parameters and, based on this, draw a conclusion to what decimal place to round (uniformly through the text).
Some verbs in the text must have uniform conjugation (see, for example, lines 112 and 119).
The sentence “significance of the models” seems incorrect. Is it a special term from statistics?
There are some typos and incorrect sentences in the manuscript – please, see the attached file (which also contains additional comments).
Kind regards!

Author Response
Editors
Catalyst
Concepción (Chile), November 30th, 2020
Dear Reviewer 2,
Please find herewith the second revised version of the manuscript titled “Optimization of Fenton technology for recalcitrant compounds and bacteria inactivation” by Pablo Salgado, José Luis Frontela and Gladys Vidal which was assigned reference no. catalysts-978824
We want to thank the reviewers for so carefully reading the manuscript and their valuable suggestions. All the comments enclosed were attended and the paper was carefully checked according to the editorial suggestions.
With my best regards,
Gladys Vidal

Round 3
Reviewer 1 Report
Dear Authors,
The authors revised the manuscript, for which a thank you is due. Unfortunately, the key comments from the review of the first version and the review of the second version of the article were not reflected. The work does not bring any cognitive novelties. It also does not bring any new developments in the scientific aspect.
I wish the authors a lot of scientific successes. Please note that the works should be original, with a clear novelty aspect. Of course, you have to follow the reports of other researchers, because they enrich and increase our knowledge, but you always should move forward.
Author Response
Concepción (Chile), december 7th, 2020
Dear Reviewer 1,
Please find herewith the secod evised version of the manuscript titled “Optimization of Fenton technology for recalcitrant compounds and bacteria inactivation” by Pablo Salgado, José Luis Frontela and Gladys Vidal which was assigned reference no. catalysts-978824
We want to thank the reviewers for so carefully reading the manuscript and their valuable suggestions. All the comments enclosed were attended and the paper was carefully checked according to the editorial suggestions.
Question:
The authors revised the manuscript, for which a thank you is due. Unfortunately, the key comments from the review of the first version and the review of the second version of the article were not reflected. The work does not bring any cognitive novelties. It also does not bring any new developments in the scientific aspect. I wish the authors a lot of scientific successes. Please note that the works should be original, with a clear novelty aspect. Of course, you have to follow the reports of other researchers, because they enrich and increase our knowledge, but you always should move forward.
Answer:
Thank you very much for all the suggestions provided. We value your time and opinion that you have given to this work and will consider your insights and advice for future jobs. We regret not being able to do anything else in this, but the COVID-19 situation does not allow us to be in our workplace, to repeat, re-think experiments, according to your suggestions.
Looking forward to hearing from you,
Yours sincerely,
Prof. Gladys Vidal
Engineering and Environmental Biotechnology Group
Environmental Science Faculty & Center EULA-Chile
Universidad de Concepción
Concepción, Chile.
E-mail: glvidal@udec.cl
Reviewer 2 Report
Dear Authors,
thank you for the work done with the manuscript to improve its quality. However, some minor corrections are still required (see attached pdf-file).
My main recommendation is:
Line 387 (Conclusions) please add some sentences about the agreement of the predicted data with the experimental results.
Final conclusion of the reviewer - the manuscript can be accepted for publication in Catalysts after adding of these corrections (see above and the attached pdf-file).
Kind regards!

Author Response
Dear Reviewer,
Attached you can find the answers to your questions/suggestions.
With my best regards,
Gladys Vidal
